# CountLoop: Training-Free High-Instance Image Generation via Iterative Agent Guidance

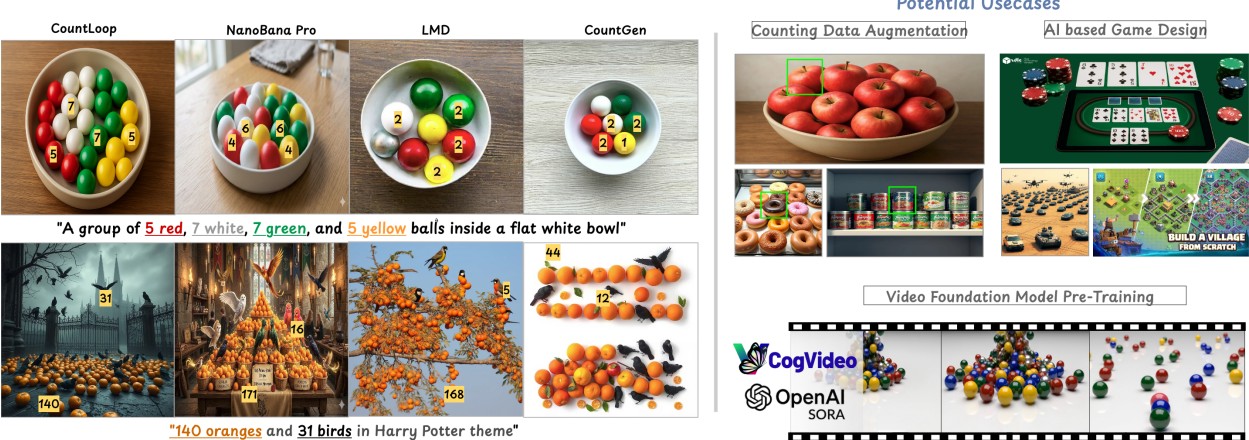

Figure 1: Given prompts with explicit per-class counts, CountLoop produces images whose detected counts align with targets, even at extreme cardinalities (*e.g.*, 140 oranges and 31 birds), where competing methods suffer from count saturation, semantic leakage, and grid-like layouts. Unlike prior approaches, CountLoop requires no retraining: a VLM-guided planning graph structures the layout, instance-driven attention masking prevents attribute leakage, and a Critic VLM iteratively refines the scene until the count and quality criteria are met. Accurate count-faithful synthesis unlocks practical applications (right): (a) augmenting object-counting datasets (Ranjan et al. (2021)) with high-instance scenes, (b) populating AI-driven games (Microsoft Research Blog (2025)) with precise entity counts critical for gameplay balance, and (c) enriching video foundation model pre-training (Wan et al. (2025); Hong et al. (2022))with diverse, numerically reliable synthetic data.

## Abstract

Diffusion models excel at photorealistic synthesis but struggle with precise object counts, especially in high-density settings. We introduce CountLoop, a training-free framework that achieves precise instance control through iterative, structured feedback. Our method alternates between synthesis and evaluation: a VLM-based planner generates structured scene layouts, while a VLM-based critic provides explicit feedback on object counts, spatial arrangements, and visual quality to refine the layout iteratively. Instance-driven attention masking and cumulative attention composition further prevent semantic leakage, ensuring clear object separation even in densely occluded scenes. Evaluations on COCO-Count, T2I-CompBench, and two newly introduced high instance benchmarks show that CountLoop reduces counting error by up to 57% and achieves the highest or comparable spatial quality scores across all benchmarks, while maintaining photorealism.

# 1   Introduction

Digital creators, designers, and artists increasingly use text-to-image diffusion models like DALL-E 3 (Betker et al. (2023)), SDXL (Podell et al. (2024)), and FLUX (Black-Forest-Labs (2024)) to produce high-quality visuals. However, these models struggle with scenes containing many distinct yet related object instances (Paiss et al. (2023)), limiting their effectiveness in applications where cardinality is crucial, such as game asset generation (*e.g.*, crowds of characters or repeated environmental elements) or augmenting object-counting datasets and even as a pretraining task in video diffusion models (Wan et al. (2025)). Current image diffusion models typically saturate at around 10 instances per category (Binyamin et al. (2024)), with precise quantity being a known long-tail compositional failure (Ye et al. (2025)), yielding semantic drift (mixed attributes), spatial collapse (cluttered or overlapping objects), or instance duplication. For instance, a prompt like "140 oranges and 31 birds in Harry Potter theme" might under/over-produce an incoherent pile of either oranges or birds or both (fig. 1), compromising accuracy and usability.

Current solutions fall into three categories: (1) text-to-image (T2I) models, sometimes augmented with gradient-based counting guidance (Kang et al. (2025); Chefer et al. (2023)); (2) layout-to-image (L2I) pipelines (Li et al. (2023); Feng et al. (2023); Binyamin et al. (2024); Zhou et al. (2024); Wang et al. (2024a); Zhou et al. (2025)); and (3) agentic diffusion frameworks (Wu et al. (2024b); Wang et al. (2024b); Yang et al. (2024); Wu et al. (2025)). However, none scale effectively to high-instance scenes or fully resolve the failure cases illustrated in fig. 2. Gradient-guided methods inject counting signals during denoising but often introduce artifacts or worsen semantic leakage as object density increases (Dahary et al. (2024; 2025)) (see fig. 2(b)). L2I pipelines guide diffusion using bounding boxes or masks, but single-pass generation causes cross-attention leakage, and autoregressive layout biases (Xiong et al. (2024)) produce unnatural, grid-like arrangements (see fig. 2(a)). Agentic frameworks use LLM-based critique but lack explicit scene structure, leading to overcorrection or object omission, and their focus on aesthetics over spatial precision makes them unreliable for dense, count-sensitive generation. We present COUNT-LOOP, a training-free framework that treats high-instance image generation as an iterative design process.

Inspired by how human designers refine compositions, COUNTLOOP parses the input prompt into a planning graph encoding object attributes and spatial relationships, which guides layout-conditioned image synthesis. A VLM critic then evaluates (a) spatial coherence and appearance fidelity via a pretrained encoder (Wu et al. (2024a)), and (b) counting accuracy via an off-the-shelf detector — since VLMs alone struggle with precise counting in dense scenes (Gavrikov et al. (2025)). Critic feedback updates the planning graph and prompt, repeating until quality criteria are met.

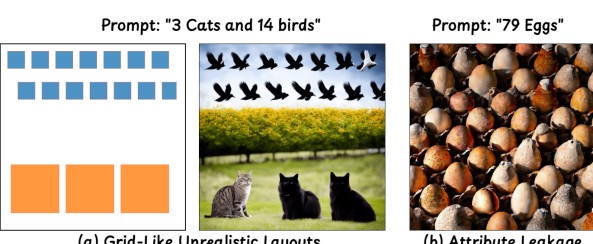

Figure 2: Issues in High-instance image generation

Our COUNTLOOP also introduces a cumulative attention mechanism during the denoising process to mitigate semantic leakage (Dahary et al. (2024; 2025)), a common issue in high-instance scenes. Inspired by the multi-turn image generation (Cheng et al. (2024)), rather than generating all subjects simultaneously, it synthesises one instance at a time, providing per-instance grounding that prevents semantic entanglement and maintains the identity of individual objects. By imposing attention locality within instance-specific regions (Chefer et al. (2023); Dahary et al. (2024)), COUNTLOOP encourages independence across objects and prevents the borrowing of features from nearby or similar instances. Together, this iterative agent-guided loop, the use of per-instance cumulative attention composition, and VLM-based visual feedback form a training-free closed-loop pipeline. Unlike gradient-guided methods that introduce artifacts or L2I pipelines that produce rigid layouts, COUNTLOOP acts as a plug-and-play enhancement to standard diffusion backbones, scaling gracefully to dense, high-instance scenes while maintaining accurate counts and natural spatial arrangements.

Our contributions: **(1)** COUNTLOOP, a training-free iterative pipeline for high-instance generation with precise counts and strong aesthetics; **(2)** a cumulative attention mechanism that sequentially injects objects via instance-specific masks, mitigating semantic leakage and preserving identity in dense scenes; **(3)** a VLM

critic that evaluates count consistency and appearance fidelity, providing structured feedback to iteratively refine layout and prompt; **(4)** evaluation on COCO-Count, T2I-CompBench, and two new high-instance benchmarks shows CountLoop reduces counting error by up to 57% on standard benchmarks and 43–48% on high-instance scenes, with the highest or comparable spatial quality across all four.

## 2 Related Work

**Count Control in Text-to-Image Generation:** Modern text-to-image diffusion models such as LDM (Rombach et al. (2022)), Imagen (Saharia et al. (2022)), SDXL (Podell et al. (2024)), and FLUX (Black-Forest-Labs (2024)) achieve high photorealistic fidelity through iterative denoising, but break down when prompts demand structured control, such as "40 red cans on a shelf" or "12 apples in a bowl and 8 on the table". Beyond 10-15 identical objects, they often miscount, exhibit attribute leakage, and suffer spatial collapse (Chefer et al. (2023); Dahary et al. (2024); Binyamin et al. (2024)). These limitations stem from architectural constraints: cross-attention fails to preserve per-instance identity, and there is no global mechanism enforcing cardinality or spatial coherence. Gradient-guided corrections (Kang et al. (2025); Zeng et al. (2025)) offer partial remedies at inference time: Counting Guidance (Kang et al. (2025)) steers denoising via a regression-based counting network, while YOLO-Count (Zeng et al. (2025)) introduces a differentiable cardinality map for token-level optimisation, improving count. However, these methods treat counting as a global scalar constraint by optimising a signal that tells the model how many objects to produce but not where to place them or how to keep them visually distinct. As density grows, this leads to object merging and spatial collapse, fixing which requires explicit layout structure and per-instance attention control rather than stronger counting gradients.

**Layout-to-Image Generation:** Layout-to-image methods condition diffusion on boxes or masks (Li et al. (2023)), LLM-derived layouts (Lian et al. (2023); Feng et al. (2023)), or per-instance conditioning signals such as instance-decomposed cross-attention with shading aggregation (Zhou et al. (2024)) and flexible bbox/point/scribble inputs (Wang et al. (2024a)). Scene-graph pipelines (Johnson et al. (2018)) encode pairwise relations but depend on expensive graph annotations. More recent approaches decouple layout planning from rendering via intermediate depth-map synthesis (Zhou et al. (2025)), improving spatial coherence across diverse backbones, while retrieval-based layout adaptation (Binyamin et al. (2024)) avoids manual annotation but depends on retrieval coverage and the downstream generator. However, shared limitations persist: single-pass generation causes cross-attention leakage and identity confusion as objects crowd together (Chefer et al. (2023); Dahary et al. (2024)), and the absence of closed-loop feedback means counting errors are irreversible. Robustness under high-instance prompts ($\gg$20) remains under-explored (Binyamin et al. (2024)).

**Agentic Diffusion Correction:** Recent frameworks employ LLM/VLM agents as planners or critics to iteratively refine diffusion generation. SLD (Wu et al. (2024b)) applies LLM-directed latent-space corrections (addition, deletion, repositioning) but lacks a persistent scene representation, limiting global spatial consistency. GenArtist (Wang et al. (2024b)) uses MLLM-driven tree planning over specialist tools, improving compositionality but not targeting high-instance count control. RPG-DiffusionMaster (Yang et al. (2024)) decomposes prompts via chain-of-thought and applies regional diffusion, but rectangular non-overlapping partitions preclude dense or occluded multi-instance layouts. Qwen-Image (Wu et al. (2025)) combines a VLM backbone with a diffusion transformer for strong semantic fidelity, yet lacks explicit layout conditioning and iterative correction. Despite advances in compositional control, none maintain a structured, editable scene graph, reducing reliability when precise counting and spatial coherence are required at high instance densities.

## 3 CountLoop

**Overview:** CountLoop is a training-free, VLM-guided framework for high-instance image generation operating in three stages (see fig. 1). First, a Design VLM interprets the prompt to produce realistic, non-grid layouts (fig. 2(a)) with natural object placement. Second, a cumulative attention mechanism guides style-consistent generation, mitigating attribute leakage (fig. 2(b)) and preserving object clarity under overlap.

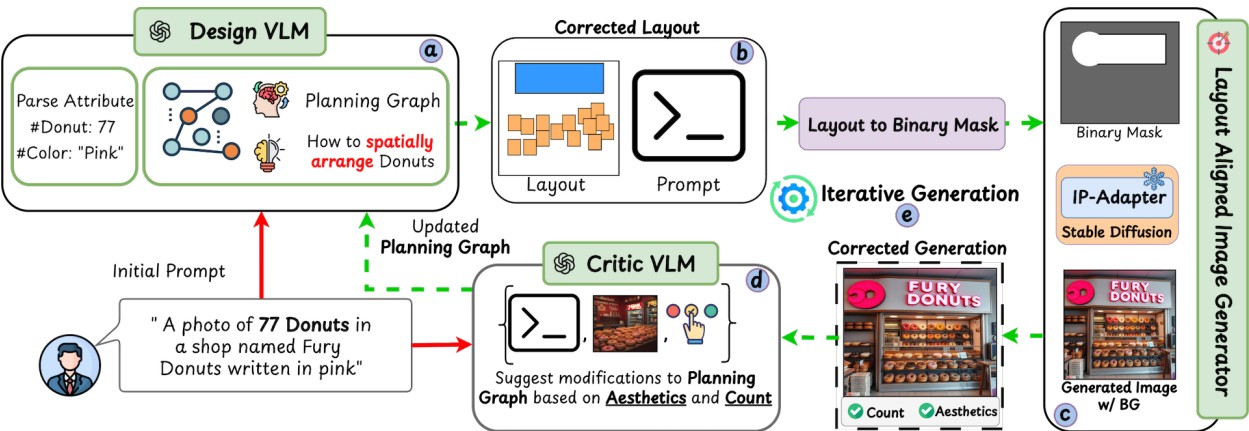

Figure 3: Given a text prompt, ⓐ The Design VLM parses the prompt to construct a planning graph, which is converted into a pixel-aligned layout ⓑ. ⓒ This layout guides an IP-Adapter-enhanced T2I backbone for image generation. ⓓ A Critic VLM evaluates the generated image's count and aesthetics, providing structured feedback to update the planning graph. ⓔ This iterative loop continues until objectives are met.

Finally, a Critic VLM assesses counting accuracy and aesthetic quality, providing structured feedback to refine both layout and prompt.

## 3.1 VLM-Guided Layout Generation

Precise multi-instance layout generation remains challenging: LLM-based approaches (Lian et al. (2023)) suffer from limited spatial reasoning (Ramachandran et al. (2025)) and autoregressive bias, producing rigid grid-like structures (fig. 2(a)), while VLMs (Wu et al. (2023a)) offer richer multimodal reasoning but still lack sufficient spatial flexibility. We address this by augmenting VLM Chain-of-Thought with explicit relational and spatial priors via planning graphs (Chen et al. (2024)). Building on Qwen3-VL (Yang et al. (2025)), our Design VLM produces more consistent object placement, attributes, and relations, reducing grid artifacts and yielding realistic compositions.

**Prompt Parsing:** As a precursor to our process, we break down the input prompt into its core components, including object-level quantities, instance-level attributes, and instance-level quantities. For example, the prompt "two cats and a bird in the sky" contains two objects, "cat" and "bird", with desired quantities of two and one, respectively. The object "bird" is associated with an instance-level attribute "in the sky", which has a desired quantity of one, whereas the object "cat" is not associated with any instance-level attributes. We begin by instructing a VLM (*e.g.*, Qwen3-VL (Yang et al. (2025))) to analyze the prompt and return a JSON dictionary. Each node carries `id`, `category`, `pos [x,y]`, `size [w,h]`, `depth`, and `color`; edges encode `relation`, `dist`, and `angle`; a `context` field captures the background. These object-attribute relations serve as the foundation for the planning graph. The full prompt schema and a worked example are provided in the Supplementary.

**Planning Graph Construction:** The graph construction process begins by using object-attribute relations parsed from the input prompt. Specifically, the planning graph is defined as $G = (V, E, B_{\text{bg}})$, where $V$ denotes object-instance nodes, $E$ represents edges encoding spatial relations, and $B_{\text{bg}}$ captures the scene context (*e.g.*, "outdoor environment"). Each node in $V$ includes attributes like category (*e.g.*, cat, bird), a unique identifier (*e.g.*, `cat_1`), normalized position $[x, y] \in [0, 1]^2$, size $[w_i, h_i]$, depth prior $d \in [0, 1]$, and color. Edges in $E$ encode spatial relations via directional operators (*e.g.*, "above," "left-of"), normalized distances, and angular orientations. $G$ enforces structured spatial reasoning, nodes specify individual properties while edges ensure relational consistency (*e.g.*, minimum distances to prevent overlaps), enabling realistic multi-object scene construction. To integrate this structured representation into VLM reasoning, we convert the graph into a textual prompt template $P_G$:

$$P_G = \phi(['Object'], ['Relation'], ['Context']) \tag{1}$$

where $\phi$ denotes a text concatenation operator; 'Object' $\in V$, 'Relation' $\in E$, and 'Context' $\in B_{bg}$ denotes the textual attributes from the planning graph. Full prompt details are provided in the supplementary. The prompt $P_G$ encodes object positions, depth, and sizes in text, enabling spatial reasoning within the VLM, combined with in-context examples for grounding. Both $P_G$ and the in-context examples (denoted by $P_{\text{icl}}$) are fed into the Design VLM as follows:

$$\mathbb{J} = \text{VLM}(P_G, P_{\text{icl}}) \tag{2}$$

where $\mathbb{J}$ is the VLM's output in JSON format, from which we extract per-instance layouts $l_i = (x_i, y_i, w_i, h_i)$ forming the layout set $\mathbb{L} = \{l_1, \ldots, l_N\}$, the scene description prompt $P_d$, and background prompt $P_{\text{bg}}$.

## 3.2 Layout Aligned Image Generation

Given layouts $\mathbb{L}$, layout-grounded generation commonly exhibits attribute leakage (Dahary et al. (2024; 2025)), yielding correct counts but degraded visual quality (fig. 2(b)). Inspired by multi-turn generation (Cheng et al. (2024)), we avoid synthesising all instances in a single pass.

**Layout Aligned Attention Masking:** Given the object layouts $\mathbb{L}$ and prompt description $P_d$, we aim to ground the layout with the text to generate images with accurate instance counts. Since layouts are discrete spatial arrangements, we project them into a continuous space using a layout encoder. Following (Lian et al. (2023)), we adapt GLIGEN(Li et al. (2023)) adapter (denoted by $\mathbb{E}$), which encodes each per-instance layout boxes $l_i \in \mathbb{L}$ into latent tokens $Q_i = \mathbb{E}(l_i)$ where $Q_i \in \mathbb{R}^D$. The full set of embeddings is represented as $Q = \{Q_1, \ldots, Q_N\}$ Since GLIGEN was built on top of SD(McLean (2023)), each layout token $Q_i$ has same dimensions $D$ as the intermediate latents of U-Net based diffusion models. These tokens are injected into the Diffusion U-Net via GLIGEN's gated self-attention mechanism in a training-free manner, conditioning the denoising process on the layouts. During denoising, the U-Net's cross-attention layers compute spatial attention between the latent feature map infused with the layout embeddings and the text embedding of $P_d$ to obtain cross-attention feature $A_{cross}$ thereby grounding the features with the textual description.

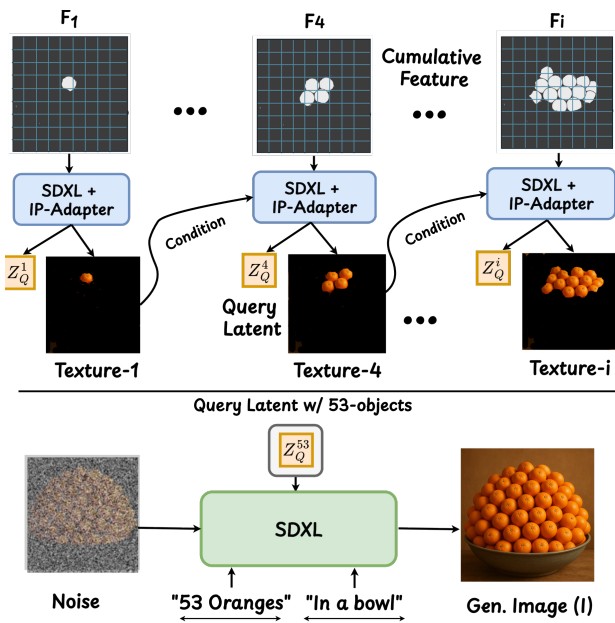

Figure 4: Cumulative latent composition, along with disentangled query feature extraction, mitigates attribute leakage.

However, directly using $A_{\text{cross}}$ for generation introduces semantic leakage (Dahary et al. (2024)) because it attempts to generate all instances at once. To mitigate this, we independently process $A_{\text{cross}}$ at the instance level. For each object instance $i$, we apply a binary spatial mask $M_i \in \{0, 1\}^{w_i \times h_i}$ (1 inside the bounding box of $l_i$, 0 elsewhere), derived from the layout $l_i \in \mathbb{L}$. The mask is then reshaped into $\hat{M}_i$ using bilinear interpolation to match the latent dimension of $A_{\text{cross}}$.

To obtain shape-aware instance boundaries, we refine $\hat{M}_i$ via the self-segmentation of (Dahary et al. (2024)), partitioning the mask into foreground/background via $k$-means ($k=2$).

This produces a binary, shape-aware mask that tightly follows object contours rather than bounding-box boundaries. The masked layout feature is then computed as:

$$A_{\text{mask}}^i = A_{\text{cross}}^i \odot \hat{M}_i \tag{3}$$

Here, $A^i_{\mathrm{mask}}$ denotes the instance-specific masked attention feature, which confines the receptive field of attention to the corresponding object's spatial region, preventing feature mixing and semantic leakage across instances.

**Cumulative Latent Composition:** Once instance-level attention maps $A^i_{mask}$ have been computed for each object layout $l_i \in L$, we build the global latent feature map $F$ by sequentially placing each object's latent features in the diffusion latent space. We initialize $F_0 = 0$ as a zero tensor in $\mathbb{R}^{H_\ell \times W_\ell \times D}$ where $H_\ell, W_\ell$ are the spatial feature dimensions and $D$ is the fixed feature dimension. Hence for $i = 0, 1, \ldots$ we update

$$F_{i+1}(x, y) = \mathbb{1}_{(x,y) \in l_i} \odot A^i_{mask} + (1 - \mathbb{1}_{(x,y) \in l_i}) \odot F_i \tag{4}$$

Here $\mathbb{1}_{(x,y)}$ is the binary indicator that pixel $(x, y)$ lies within the spatial extent of $l_i$. In other words, for each pixel covered by layout $l_i$, we replace the previous feature with the new attention feature $A^i_{mask}$, and elsewhere we retain the existing feature. Because we never change the feature dimensionality in this process (each $F_i$ and $A^i_{mask} \in \mathbb{R}^D$), the dimension $D$, remains fixed (e.g. $D = 1280$), ensuring compatibility with a frozen backbone. All object positioning, scale, and depth ordering are pre-specified by the layout $l_i$; we apply no further latent-space warping or scaling. In practice, instances are composed in order of increasing depth (Far $\rightarrow$ Near), so that each nearer object $i$ overwrites any existing features in its mask.

The result is a composite latent feature map $F$ that faithfully encodes each object's appearance, position, and scale according to the input layouts, with nearer objects occluding farther ones without any spurious blending.

**Appearance Consistency via IP-Adapter:** Generating images independently from disentangled features $F$ reduces semantic leakage but often introduces texture inconsistency, since each latent $F_i$ is denoised separately. To counter this, we condition the diffusion model (*e.g.*, SDXL (Podell et al. (2024))) on the foreground texture of the previously generated output using IP-Adapter (Ye et al. (2023)). Because leakage occurs when query tokens attend to different instances during self-attention (Dahary et al. (2024)), we further preserve the per-instance query representation ($Z_q$) before its interaction with keys and values, maintaining instance-level semantics. Formally:

$$I_{i+1}, Z_q^{i+1} = \Phi(F_{i+1}, P_d, \theta(I_i)), \quad i = 1, \ldots, N-1 \tag{5}$$

where $I_i$ is the image generated from $F_i$, $N$ is the number of objects, and $\theta$ is IP-Adapter conditioning. The first image is generated without IP-Adapter due to the absence of prior texture. Iterating over all $F_i$ aligns prompt semantics $P_d$ with accumulated visual cues, reducing hallucinations and preserving object distinctiveness. After extracting all query embeddings $Z_q = \{Z_q^1, \ldots, Z_q^N\}$, we produce a final image with minimal attribute leakage. To generate the final composition, we use the last query latent $Z_q^N$, which encodes all $N$ objects with consistent appearance. The attention operation is defined as: $\mathbb{A}(Z_q^N, K, V)$, where $K$ and $V$ are the keys and values (see fig. 4) of the diffusion. Each object-specific feature in $Z_q^N$ attends to a shared key–value set, enforcing semantic coherence across foreground instances while keeping the background disentangled. This operates as an implicit variant of self-attention expansion in video diffusion (Wu et al. (2023b); Alimohammadi et al. (2025)), but the attention is shared across object instances rather than frames. Since using only the foreground prompt $P_d$ may yield a weak background, we concatenate a dedicated background prompt $P_{\mathrm{bg}}$ with $P_d$ as the textual condition to the model. The resulting image $I$ (see fig. 4) preserves the planned layout with semantically separated objects and reduced attribute leakage.

### 3.3 Layout Refinement via Iterative Feedback

After generating image $I$, we run an iterative refinement loop that (i) evaluates $I$, (ii) identifies flaws, and (iii) updates the planning graph and prompt until count and aesthetic targets are met.

**Critic VLM:** We reconfigure a Qwen3-VL (Yang et al. (2025)) agent as a Critic VLM that analyses generated images and suggests layout revisions. Since LLM behaviour varies sharply with instruction design (Madaan et al. (2023); Sun et al. (2023)), the same model can serve as either creator or critic depending on the prompt.

Exploiting this, we supply a critique-style prompt $P_{\text{crit}}$ to the VLM which evaluates the generated image $I$ on two aspects: (a) object count fidelity and (b) visual aesthetics, as shown in fig. 3. Since VLMs remain unreliable at dense counting (Guo et al. (2025)), we obtain the count accuracy $s_c$ from an open-vocabulary detector (Liu et al. (2024)), distinct from the evaluation detector (section 4.1). Aesthetic alignment $s_a$ is scored by an external estimator (Wu et al. (2024a)). A composite score : $S = \alpha \cdot s_c + (1 - \alpha) \cdot s_a$, where $s_c, s_a \in [0, 1]$, is used to capture the overall quality of the generation (formulation details in the Supplementary). The Critic produces structured feedback in the form of text (denoted by $P_{\text{feed}}$) which is used to update the nodes and relations of the planning graph $P_G$, thereby altering the object layout size and spatial locations in the canvas.

**Parameter-Free Refinement:** The Critic VLM's textual feedback must be translated into concrete edits to the planning graph to generate an updated image incorporating the feedback. Instead of fine-tuning model parameters, we employ a parameter-free textual refinement operator inspired by (Yuksekgonul et al. (2025)). We denote this operator as $\Psi$,

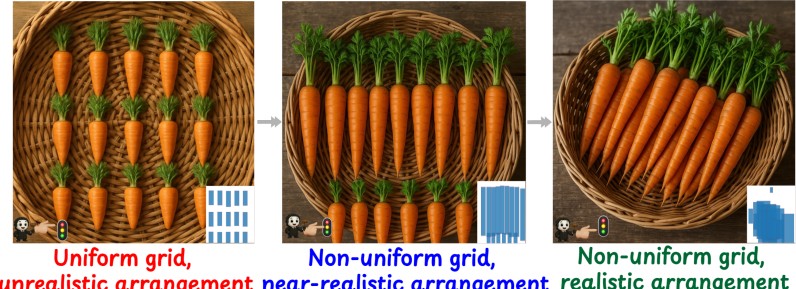

Uniform grid, unrealistic arrangement — Non-uniform grid, near-realistic arrangement — Non-uniform grid, realistic arrangement

Figure 5: Successive layout refinement by Critic VLM. Layouts in the inset.

an LLM-based text-editing agent that updates the planning graph through structured natural-language reasoning. Given the current graph $G$, the critic feedback $P_{\text{feed}}$, and an optimisation prompt $P_{\text{opt}}$, the operator produces an updated graph: $G' = \Psi(G, P_{\text{feed}}, P_{\text{opt}})$. Mirroring how PyTorch's AutoGrad (Paszke et al. (2017)) performs gradient updates, $\Psi(\cdot)$ interprets the input feedback and estimates a textual analogue of a *gradient*, using a loss function defined as a pre-defined textual prompt template in $P_{\text{opt}}$. It then applies gradient-like edits to $G$ via textual modifications rather than numerical parameter updates.

Operating entirely on textual representations, $\Psi$ applies targeted structural edits to $G$. For example: ① For feedback such as "$cup_7$ `overlaps with` $cup_3$", it increases spatial separation in $G$. ② For "`only 28 cups detected but target is 30`", it inserts the missing object nodes. This parameter-free refinement is compatible with any frozen diffusion model. After obtaining $G'$, we derive $P_{G'}$ ( eq. (1)) to generate a refined layout $\mathbb{L}$ (eq. (2)), followed by updated image synthesis $I$ (see fig. 5). The process terminates when the composite score exceeds a quality threshold, *i.e.*, the detector confirms the correct count and the aesthetic score is acceptable, or after a fixed number of rounds to prevent diminishing returns from further refinement. In practice, the majority of prompts converge within three rounds (see convergence analysis in the Supplementary).

## 4 Experiments

### 4.1 Dataset and Evaluation

**Datasets and Metric:** We evaluate on four sets spanning instance count and compositional difficulty: COCO-Count (MS-COCO subset (Lin et al. (2014))); T2I-CompBench Count (subset of (Huang et al. (2023))); newly proposed COUNTLOOP-S (single category, 200 prompts, 30–200 instances); and COUNTLOOP-M (multi-category, 200 prompts, 30–200 instances). Benchmark construction details and prompt lists are in the supplementary. We report *counting accuracy* using MAE metric (Binyamin et al. (2024)) where OWLv2 (Minderer et al. (2023)) is used as an evaluator for *all* methods; any systematic detection bias at high densities therefore affects baselines and COUNTLOOP equally, leaving relative MAE comparisons valid. Spatial alignment is measured via CLIP–FlanT5 encoder from VQAScore (Li et al. (2024)).

**Competitors:** We compare COUNTLOOP with representative T2I (SDXL (Podell et al. (2024)), FLUX (Black-Forest-Labs (2024)), SDXL-Turbo (Sauer et al. (2024)), SD3.5 (Stability-AI (2025)), Counting Guidance (Kang

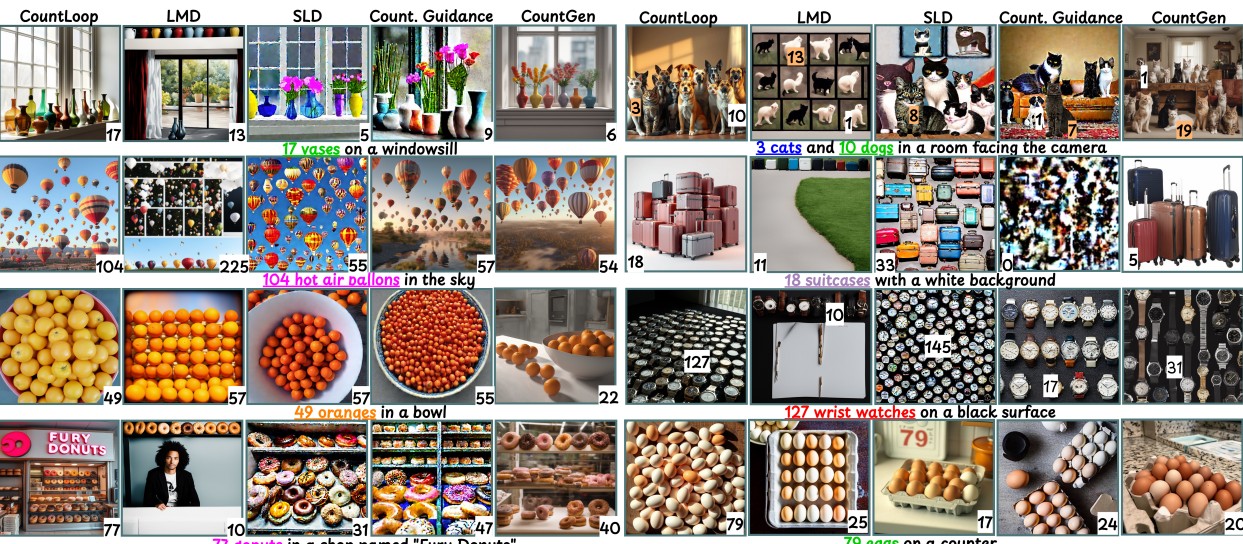

Figure 6: CountLoop maintains precise object counts and natural arrangements in dense scenes, while methods like LMD (Lian et al. (2023)), SLD (Wu et al. (2024b)), Counting Guidance (Kang et al. (2025)), and CountGen (Binyamin et al. (2024)) exhibit abnormal counts, spatial collapse, and grid artifacts. More visuals in the supplementary.

et al. (2025))), Agentic (Qwen-Image (Wu et al. (2025)), GenArtist (Wang et al. (2024b)), SLD (Wu et al. (2024b)), RPG-DiffusionMaster (Yang et al. (2024))), and L2I (LMD (Lian et al. (2023)), MIGC (Zhou et al. (2024)), CountGen (Binyamin et al. (2024)), 3DIS (Zhou et al. (2025)), InstanceDiffusion (Wang et al. (2024a))) methods. Implementation details are provided in the supplementary.

## 4.2 Main Results

**Quantitative Results:** table 1 reports counting error (MAE, lower is better) and spatial quality across all benchmarks. In the low-instance regime of COCO-Count, CountLoop achieves the lowest overall MAE (0.45), outperforming strong agentic competitors such as Qwen-Image (1.04) and SLD (1.15). These gains are modest in absolute terms because existing methods already perform well at low counts, where the failure modes CountLoop targets, like semantic leakage, layout rigidity, and count saturation, have not yet manifested. The critical differentiator emerges at scale: on CountLoop-S, CountLoop records a MAE of 7.59, less than half the error of the strongest agentic competitor (Qwen-Image: 17.30) and the best L2I baseline (3DIS: 14.55). Methods that perform competitively at low counts suffer clear collapse at scale: CountGen's MAE rises from 1.88 to 34.44 and SLD's from 1.15 to 29.65. Even recent baselines evaluated only on CountLoop-S (InstanceDiffusion: 16.07, Qwen-Image: 17.30) remain 2× above CountLoop. This robustness extends to multi-category scenes (CountLoop-M, MAE: 2.13). Notably, CountLoop achieves this without sacrificing generation quality: its spatial score on CountLoop-S is 0.93 versus 0.75 (SLD) and 0.74 (Qwen-Image), demonstrating that the Critic VLM resolves the count-quality trade-off that constrains all prior paradigms.

**Qualitative Results:** fig. 6 demonstrates CountLoop's consistent precision across diverse instance counts. For "17 vases", competitors under-generate (LMD: 13, Count Guidance: 9, CountGen: 6), while CountLoop accurately renders all 17 with natural arrangements. In the "104 hot air balloons" scene, CountLoop precisely places all balloons with realistic spacing, unlike Count Guidance (57), CountGen (54), and LMD's artificial clusters (225 overlapping). CountLoop consistently avoids semantic drift, grid artifacts, and count inaccuracies that plague baselines, outperforming all competitors on high-instance scenes.

Table 1: **Counting and spatial quality across benchmarks.** We report counting error (MAE↓) and spatial quality (Spatial↑) for single-category and multi-category prompts. **Best** in bold, second-best underlined.

| Family | Model | Single Category | | | | | | Multi Category | |
| | | COCO-Count | | T2I-CompBench | | CountLoop-S | | CountLoop-M | |
| | | MAE↓ | Sp.↑ | MAE↓ | Sp.↑ | MAE↓ | Sp.↑ | MAE↓ | Sp.↑ |
|---|---|---|---|---|---|---|---|---|---|
| **T2I** | SDXL (Podell et al. (2024)) | 2.37 | 0.38 | 2.72 | 0.75 | 29.96 | 0.63 | 9.89 | 0.55 |
| | FLUX (Black-Forest-Labs (2024)) | 1.40 | 0.53 | 1.48 | 0.78 | 17.47 | 0.65 | 9.62 | 0.58 |
| | SD 3.5 (Stability-AI (2025)) | 1.10 | 0.46 | 1.58 | 0.76 | 21.81 | 0.64 | 8.40 | 0.56 |
| | SDXL-Turbo (Sauer et al. (2024)) | 2.50 | 0.23 | 3.76 | 0.53 | 51.14 | 0.39 | 9.95 | 0.37 |
| | Counting Guidance (Kang et al. (2025)) | 1.68 | 0.63 | 3.90 | 0.56 | 42.49 | 0.47 | 8.43 | 0.41 |
| **L2I** | LMD (Lian et al. (2023)) | 3.09 | 0.24 | 5.56 | 0.73 | 16.62 | 0.66 | 6.34 | 0.64 |
| | MIGC (Zhou et al. (2024)) | 1.83 | 0.36 | 2.96 | 0.65 | 17.54 | 0.65 | 6.28 | 0.62 |
| | CountGen (Binyamin et al. (2024)) | 1.88 | 0.61 | 5.22 | 0.75 | 34.44 | 0.72 | 6.46 | 0.69 |
| | InstanceDiffusion (Wang et al. (2024a)) | 1.77 | 0.40 | 2.83 | 0.68 | 16.07 | 0.74 | 6.11 | 0.66 |
| | 3DIS (Zhou et al. (2025)) | 1.56 | 0.42 | 2.56 | 0.70 | 14.55 | 0.76 | 5.75 | 0.69 |
| **Agentic** | GenArtist (Wang et al. (2024b)) | 1.50 | 0.45 | 1.50 | 0.70 | 32.47 | 0.60 | 4.93 | 0.57 |
| | SLD (Wu et al. (2024b)) | 1.15 | 0.70 | 1.44 | 0.77 | 29.65 | 0.75 | 3.74 | 0.65 |
| | RPG (Yang et al. (2024)) | 1.28 | 0.60 | 1.47 | 0.75 | 31.85 | 0.70 | 4.34 | 0.62 |
| | Qwen-Image (Wu et al. (2025)) | 1.04 | 0.58 | 1.26 | 0.77 | 17.30 | 0.74 | 6.92 | 0.66 |
| **Ours** | **CountLoop** | **0.45** | **0.93** | **1.23** | **0.79** | **7.59** | **0.93** | **2.13** | **0.73** |

## 4.3 Ablations and Analysis

**Key Components:** table 2b progressively builds CountLoop from a plain baseline to quantify the contribution of each component on CountLoop-S. Both the Planning Graph (**PG**) and Cumulative Attention (**CA**) independently halve the counting error relative to the baseline (MAE: $29.91 \rightarrow 14.98$ and $14.39$, respectively), confirming that structured prompt decomposition and leakage-free attention each address a distinct and roughly equal source of failure. Combining them (PG+CA: 11.27) yields further gains, and Iterative Refinement (**IR**) closes the remaining gap with a 33% additional MAE reduction (11.27 → 7.59), recovering instances that a single forward pass cannot place correctly. We also show that beyond a critical instance count, the physical area per instance on a fixed-resolution canvas becomes too small for objects to remain visually distinguishable, which is a limit shared by all generation methods and that CountLoop reaches this floor at substantially higher $N$ than competitors (fig. 7a).

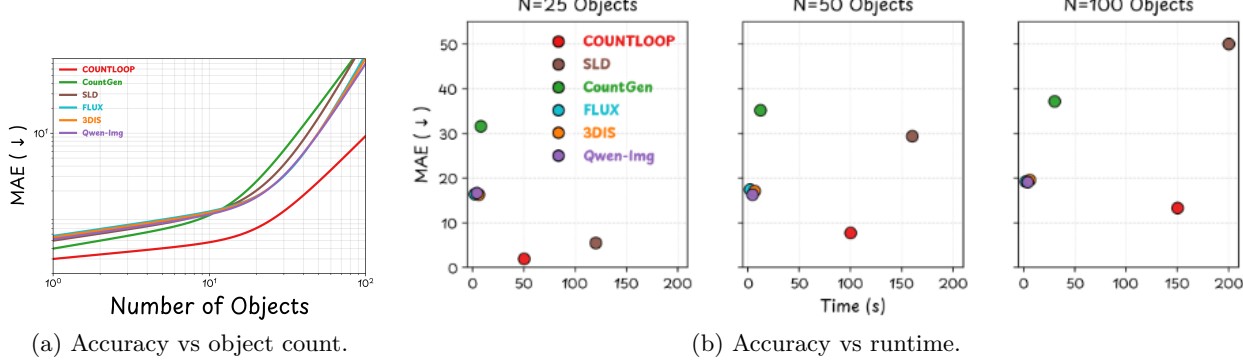

(a) Accuracy vs object count.          (b) Accuracy vs runtime.

Figure 7: *Left:* Counting difficulty rises with instance count. *Right:* Runtime curves echo the same ordering.

**Runtime Analysis:** We evaluate end-to-end runtime on CountLoop-S by measuring MAE vs. wall-clock time at $N \in \{25, 50, 100\}$ (fig. 7b). One-shot baselines (FLUX (Black-Forest-Labs (2024)), Qwen-Image (Wu et al. (2025))) are fast ($< 10\,\text{s}$) but incur high, irreducible error. L2I methods invest moderate compute yet plateau at comparable error. Among iterative methods, CountLoop dominates SLD (Wu et al. (2024b)) at every scale: at $N=100$, $3\times$ lower error (MAE $\approx 13$ vs. 50) in 25% less time ($\sim150\,\text{s}$ vs. $200\,\text{s}$); at $N=25$, MAE $\approx 5$ in $\sim50\,\text{s}$ while SLD needs $\sim125\,\text{s}$ yet plateaus at MAE $\approx 18$. This mirrors fig. 7a, where competing methods plateau beyond $\sim$10–20 objects.

Table 2: Analysis of CountLoop components, critic choices, and human evaluation. **(a)** Design-critic combinations on CountLoop-S; the default is in **bold** and the best per designer is underlined. **(b, c)** Ablations of the design components (**PG**: Planning Graph, **CA**: Cumulative Attn., **IR**: Iterative Refinement) and critic modules (**OVD**: Open-vocab Detector, **AS**: Aesthetic Scorer). **(d)** User scores on a 0–5 scale (higher is better).

(a) Design–critic configurations

| Design | Critic | MAE↓ | Spatial↑ |
|---|---|---|---|
| Qwen3-VL (Yang et al. (2025)) | Qwen3-VL | **7.59** | **0.93** |
| | Llava-1.5B | 12.40 | 0.70 |
| | Pixtral | 11.83 | 0.72 |
| Llava-1.5B (Lin et al. (2024)) | Qwen3-VL | 11.27 | 0.74 |
| | Llava-1.5B | 10.85 | 0.73 |
| | Pixtral | 10.51 | 0.75 |
| Pixtral (Agrawal et al. (2024)) | Qwen3-VL | 10.18 | 0.76 |
| | Llava-1.5B | 11.05 | 0.72 |
| | Pixtral | 10.68 | 0.73 |

(b) Component ablation

| PG | CA | IR | MAE↓ | Spatial↑ |
|---|---|---|---|---|
| ✗ | ✗ | ✗ | 29.91 | 0.61 |
| ✓ | ✗ | ✗ | 14.98 | 0.68 |
| ✗ | ✓ | ✗ | 14.39 | 0.71 |
| ✓ | ✓ | ✗ | 11.27 | 0.81 |
| ✓ | ✓ | ✓ | **7.59** | **0.93** |

(c) Critic VLM configs

| OVD | AS | MAE↓ | Spatial↑ |
|---|---|---|---|
| ✗ | ✗ | 29.59 | 0.67 |
| ✗ | ✓ | 15.49 | 0.70 |
| ✓ | ✗ | 9.27 | 0.83 |
| ✓ | ✓ | **7.59** | **0.93** |

(d) User evaluation

| Metric | CountLoop | LMD | FLUX | SLD | CountGen |
|---|---|---|---|---|---|
| Alignment | **4.5** | 3.4 | 3.7 | 4.0 | 3.6 |
| Aesthetics | **4.4** | 3.3 | 3.5 | 3.9 | 3.8 |
| Count | **4.6** | 3.7 | 4.0 | 4.2 | 3.4 |
| Overall | **4.5** | 3.5 | 3.7 | 4.0 | 3.6 |

**Performance with different Design-Critic variants:** We evaluate the impact of various Design–Critic configurations on CountLoop-S, pairing three open-source Design VLMs with three Critic VLMs under matched evaluation conditions. Results are in table 2a. The default Qwen3-VL (as design and critic role) achieves the best performance. However, even the weakest configuration (Qwen3-VL+Llava-1.5B) still outperforms the best competitor 3DIS(Zhou et al. (2025)) which is training based, demonstrating that CountLoop is robust to VLM choice while successful in mutually guiding each other to generate plausible images.

**Critic Composition:** table 2c isolates each Critic component. A VLM-only critic yields the weakest performance (MAE 29.59), confirming VLMs struggle with dense counting (Guo et al. (2025)). AS alone substantially reduces MAE to 15.49 ($-14.1$), while OVD alone achieves a larger gain (MAE 9.27, $-20.3$), identifying it as the primary driver of count accuracy. Together they reach MAE 7.59, showcasing the importance of OVD and AS in guiding our critic-VLM.

**Human Evaluation:** We ran a 30-participant study (20 designers, 10 AI artists) across all four benchmarks. Each participant rated 15 blinded sets of 5 images (CountLoop, FLUX (Black-Forest-Labs (2024)), LMD (Lian et al. (2023)), SLD (Wu et al. (2024b)), and CountGen (Binyamin et al. (2024))) on a 5-point scale for Prompt Alignment, Aesthetic Quality, Count Accuracy, and Overall Preference. CountLoop was preferred across all axes (table 2d), with clear margins over its competitors. Human count accuracy scores are consistent with OWLv2-reported trends across all methods, providing the strongest detector-agnostic validation available: if OWLv2 introduced systematic bias at high counts, human rankings would diverge from detector rankings; they do not. Procedure, demographics, and the survey interface are detailed in the supplementary.

**Performance across different T2I backbones:** To assess the generality of CountLoop across diffusion backbones, we replaced the default SDXL model with two additional Stable Diffusion checkpoints: *SD v1.5* and *SD 3.5*. We kept all other components (planning graph, cumulative attention, IP-Adapter, critic loop) and hyperparameters identical. table 3 reports counting MAE, and spatial scores on the CountLoop-S benchmark. While all backbones benefit substantially from Count-

LOOP's structured refinement, we observe that higher-capacity models yield marginally better spatial coherence, with SDXL at the top. Counting performance remains stable (MAE $\leq 8.1$) across all three backbones, indicating that COUNTLOOP's instance-control mechanism is largely model-agnostic.

**Robustness to Critic Components:** The Critic VLM relies on two external modules: an open-vocabulary detector (OVD) for count verification and an aesthetic scorer (AS) for visual quality assessment. By default, we use the base GroundingDINO (Liu et al. (2024)) checkpoint as the OVD and Q-Align (Wu et al. (2024a)) as the AS across all experiments. GroundingDINO is selected for its strong performance on dense open-vocabulary detection, while Q-Align is chosen for its discrete text-defined level design, which produces stable scalar scores with lower variance than continuous VLM-based scoring, a critical property for a reliable termination signal in the iterative critic loop. This stability advantage is consistent with recent findings (Cao et al. (2025)). The critic detector is intentionally distinct from the evaluation detector (OWLv2 (Minderer et al. (2023))) to prevent the critic from optimising directly against the evaluation metric; we fix the GroundingDINO confidence threshold to 0.3 across all experiments for reproducibility.

Table 3: Backbone swap.

| Backbone | MAE↓ | Spatial↑ |
|----------|------|----------|
| SD v1.5 | 8.05 | 0.88 |
| SD 3.5 | 7.44 | 0.90 |
| SDXL | **7.59** | **0.93** |

To verify that COUNTLOOP's gains are driven by the iterative loop architecture rather than a specific component pairing, we swap each module independently (table 4). Replacing GroundingDINO with OWLv2 in the critic role introduces critic–evaluator overlap that gives OWLv2 an inherent advantage, yet MAE remains comparable to the default. Replacing Q-Align with ImageReward (Xu et al. (2023)) increases MAE modestly but still substantially outperforms the best external baseline (3DIS: 14.55). Even the weakest configuration in the table beats all published baselines by a wide margin, confirming that COUNT-LOOP is robust to component choice. Notably, the [†]OWLv2-as-critic row is an implicit cross-detector check: despite critic–evaluator overlap giving OWLv2 an inherent advantage, MAE remains comparable to the GroundingDINO default, confirming gains are not a detector artefact.

Table 4: Component swap on COUNTLOOP-S. Evaluation detector is OWLv2 throughout. [†]OWLv2 as both critic and evaluator gives an inherent advantage, yet MAE remains comparable.

| Critic OVD | Aesthetic | MAE↓ | vs Best Baseline |
|------------|-----------|------|------------------|
| GroundingDINO | Q-Align (default) | **7.59** | +48% |
| OWLv2[†] | Q-Align | 8.57 | +41% |
| GroundingDINO | ImageReward | 9.21 | +36% |

**Instance Count Scalability by Object Regime:** fig. 8 disaggregates the MAE-$N$ relationship by object-size regime, grouping categories into *large* (balloons, elephants, trucks), *medium* (birds, cats, oranges), and *small* (watches, buttons, roses) classes, directly addressing the question of whether COUNTLOOP has a practical upper bound on reliable instance generation.

Across all six methods, MAE increases monotonically with $N$, consistent with the aggregate trend. The rank ordering Large < Medium < Small is *method-agnostic*: it reflects two compounding factors independent of generation strategy. First, small objects occupy a larger fraction of the canvas at high instance counts, intensifying cross-attention identity confusion during diffusion sampling. Second, OWLv2 localisation precision degrades on densely packed, sub-pixel instances, introducing systematic upward bias in the detector-based MAE estimate for all methods equally. Since both factors are architectural constants of the evaluation setup rather than properties of any single method, the tier ordering cannot be attributed to COUNTLOOP's design, and the shared bias does not inflate COUNTLOOP's relative advantage, as it affects every method in the comparison identically.

Critically, COUNTLOOP maintains the lowest MAE in every regime at every $N$, with the advantage *widening* rather than narrowing at high density: the gap over next-best 3DIS reaches $\Delta$MAE $\approx 7.6$ in the small-object regime at $N = 200$. COUNTLOOP achieves 87% count accuracy in the hardest setting (small, $N = 200$) and 92% in the large-object regime, well beyond all prior methods. The performance ceiling is *category-dependent* and scales with object size; COUNTLOOP raises it substantially across all regimes, with its advantage widening at the benchmark ceiling of $N = 200$.

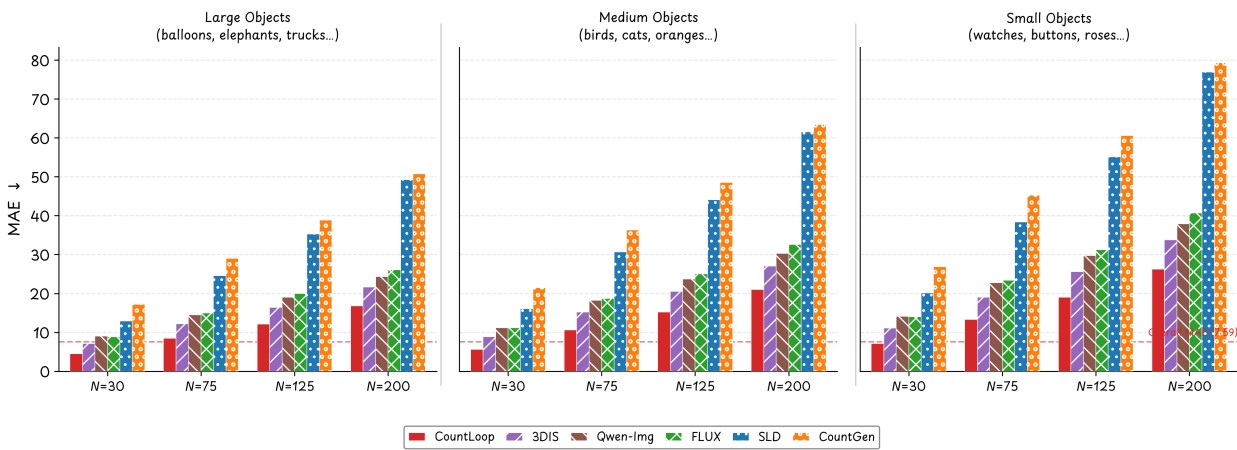

Figure 8: **Per-regime MAE at representative instance counts on CountLoop-S.** Categories are grouped by object size (Large / Medium / Small); bars show MAE at $N \in \{30, 75, 125, 200\}$ for all six methods. The Large $<$ Medium $<$ Small ordering is method-agnostic, reflecting canvas-density pressure and OWLv2 precision loss on small, densely-packed instances. CountLoop (red) achieves the lowest MAE in every regime at every $N$; the gap over next-best 3DIS widens with $N$, reaching $\Delta$MAE $\approx 7.6$ at $N = 200$ in the small-object regime (87% vs. 83% count accuracy). The dashed reference line marks CountLoop's overall MAE on CountLoop-S.

## 5    Conclusion

We presented CountLoop, a training-free, iterative framework for high-instance image generation with precise object counts and strong visual quality. VLM-based planning graphs, instance-driven attention, and cumulative attention composition overcome count saturation, semantic leakage, and rigid layouts, with a critic-in-the-loop refining generation by updating layout and prompts. On COCO-Count, T2I-CompBench, and new high-instance benchmarks, CountLoop reduces counting error by up to 57% on standard benchmarks and 43-48% on high-instance scenes, achieving the highest or comparable spatial quality throughout.

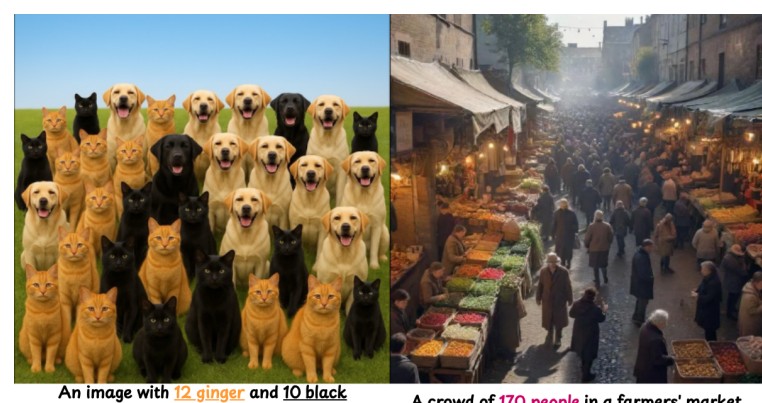

Figure 9: Failure cases

**Future Work:** It would be interesting to extend CountLoop to layout-free generation with weak spatial priors, and improve human modeling in dense scenes.

**Limitations:** As a training-free system, CountLoop inherits the limitations of its frozen VLM and detector, allowing their biases to propagate. Dense occlusions, especially in human scenes, can degrade attention quality and spatial consistency. Lacking explicit 3D priors, CountLoop struggles with generating objects in different poses and complex perspectives. Moreover, strong layout guidance can reduce intra-class diversity by biasing toward canonical poses or textures for count accuracy. Some of these limitations are shown in fig. 9. Integrating this approach with FLUX-based DiT models may yield valuable insights.

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
