# OpenReview forum: "CountLoop: Training-Free High-Instance Image Generation via Iterative Agent Guidance"
_TMLR — Under review for TMLR_

### Review · Reviewer_jkub · 2026-07-03

**Summary Of Contributions:**

CountLoop is a training-free pipeline for generating images with a specified, possibly large number of object instances. The framework: a VLM parses the prompt into a planning graph and a layout, an SDXL backbone renders instances one at a time with per-instance attention masking, and a critic loop built on a VLM plus an open-vocabulary detector edits the planning graph for up to three rounds. For experiments, the paper introduces two high-instance benchmarks covering 30 to 200 instances, reports visible MAE improvements over 14 open-source baselines, and shows that CountLoop-generated data improves a counting model on FSC-147.

**Strengths**: the ablations and robustness checks are fairly thorough, runtime and failure cases are honestly disclosed, and the FSC-147 result is evidence of practical value independent of the paper's own protocol. The high-instance regime is underexplored.

**Weaknesses**: my main concern is practical utility. Even with the full pipeline the counts remain substantially off at high instance numbers, and the count-critical applications the paper motivates itself with do not obviously tolerate such errors, so it is unclear what the method is currently accurate enough for. Beyond that, the evaluation rests on an untested assumption about detector reliability in exactly the regime the paper targets, and several claims are stronger than the evidence.

**Additional Comments:**

Per Table 2b, the planning prior and attention composition contribute most of the improvement, while the iterative loop refines the remaining step and is idle for the 60% of prompts that converge in one round. The framing emphasizes the loop, but the numbers suggest the layout prior and the composition mechanism deserve more of the credit.

**Audience:**

Yes

**Audience Explanation:**

Count control is an active problem, the 30-200 regime has essentially no prior quantitative work, and the benchmarks and the FSC-147 result would interest people working on compositional generation and synthetic training data.

**Broader Impact Concerns:**

None beyond what the supp's Broader Impact statement already covers. The statement is adequate.

**Claims And Evidence:**

No

**Claims Explanation:**

The following issues lead me to answer no.

1. The evaluation rests on an untested assumption. Every quantitative result is mediated by an open-vocabulary detector, GroundingDINO in the critic and OWLv2 in the metric, yet neither is validated as a counter in the 30-200 regime. The paper's own Supp Table 1 points the other way: on the dense-counting benchmark FSC-147, GroundingDINO reaches a test MAE of 54.16, roughly ten times worse than the dedicated counting models in the same table (LOCA at 10.79, CountGD at 5.32), so the tool that verifies counts inside the loop is itself weak at dense counting. Sec 4.1 argues the detector bias affects baselines and CountLoop equally. The paper does not test this, and the setup itself suggests otherwise: the Design VLM prompt in Supp Fig 8 constrains layouts so that instances remain detectable, so CountLoop's outputs sit where detectors work well, while the baselines' characteristic failures, grids, heavy overlap, merged instances, sit exactly where detector counts become ill-defined, and where the true count is ambiguous even for humans. The human study does not resolve this, since raters give 0-5 ratings rather than actual counts.
2. The precision claims are stronger than the results. The paper implies exactness throughout, from precise instance control in the abstract to Supp Sec 4, which states the loop corrects the scene until per-class counts match exactly, with counts guaranteed by construction. The paper's own numbers contradict this: MAE 7.59 on CountLoop-S, an average miss of roughly 26 objects in the hardest setting at N=200, and 10% of prompts never converging within three rounds. The paper also reports only MAE, while the prior works its evaluation builds on, CountGen for COCO-Count and the LMD benchmark used by SLD, report exact-match accuracy, under which every method including CountLoop would likely sit near zero at high counts. The count-critical applications in Fig 1 tolerate neither being off by 20 nor by 40, and the paper doesn't state what tolerance each application requires.
3. The application claims go beyond the experiments. Of the three applications in Fig 1, only data augmentation is validated. I understand the method is scoped to images, but the claims are not scoped accordingly: the game and video pretraining uses involve dynamics, which the framework can neither generate nor verify, and nothing in the experiments supports them. The paper's own diagnosis points the same way: Sec 2 asserts as fact that counting failures stem from architectural constraints, a claim that is debatable in itself, and if it is right, then feeding count-faithful images back into pretraining cannot teach a generator to count, and the pretraining application loses its basis.
4. Question on runtime: Sequential per-instance generation at 50 denoising steps plus refinement rounds seems hard to fit into the 150 s at N=100 reported in Fig 7b. What does that figure include?

The underlying system appears solid and the ablations are careful. My no is about the gap between what is claimed and what is shown, and about the unvalidated measurement behind the numbers. And if the paper is to deliver the downstream role it envisions for itself, the counts need to be reliable, and errors of 20 to 40 objects fall short of that.

**Requested Changes:**

Critical:

1. Validate the evaluation. Human point counts on a subset stratified by count and occlusion, with detector error reported per method, so the equal-bias assumption is tested rather than asserted. Report the OWLv2 threshold and NMS settings, currently absent, with a sensitivity sweep. State the visible-instance counting convention in the main text.

2. Calibrate the precision claims. Remove or qualify "match exactly" and "guaranteed by construction", and report exact-match rates and tolerance curves per count tier alongside MAE.

3. Reconcile the runtime with the described method and state what Fig 7b includes.

4. Calibrate the application claims to the supp's own potential-use-cases framing or add a minimal validation.

5. State what generates the input layouts for the layout-to-image baselines on the text-only benchmarks, since Table 1's fairness depends on it, and correct the metric attribution in Sec 4.1: Binyamin et al. report exact-count accuracy with YOLOv9, not MAE with OWLv2, so the protocol is the authors' own and should be presented as such.

Would strengthen the work:

6. A commercial reference point on even a subsample. Fig 1 already includes a qualitative Nano Banana Pro comparison, and published commercial counting evaluations stop near 15 objects, so this would likely strengthen the motivation.

7. Audit the label fidelity of the FSC-147 augmentation corpus, and add a density-based counter as a second evaluator, a stronger cross-check than swapping between two similar detectors.

---

### Review · Reviewer_4vx5 · 2026-07-11

**Summary Of Contributions:**

strengths:

This paper CountLoop a training-free, iterative framework for high-instance image generation. This methods leverage the test-time-scaling with the multiple VLM-refinement iterations to achieve 1) precise object counts 2) realistic arrangement and 3) strong visual quality. The experiment results on the widely-used benchmarks demonstrate the effectiveness of proposed CountLoop

weakness:

1. What is 'attribute leakage'? Is it another name for an artifact? Why does Fig. 2(b) reflect the 'attribute leakage'?
2. Parameter-Free Refinement is not illustrated clearly. I cannot get its relation to PyTorch’s AutoGrad. I don't know what the feedback message is to refine the layout from left to right in Fig. 5.
3. VLM is leveraged to give feedback on the multi-round generation, but VLM is not stable and robust enough to give a 100% accurate counting results, which is not as strong as the counting expert model.
4. Doubt on the task or potential use cases. The current CountLoop method cannot produce 100% accurate counts. It is implausible that we further apply this technique as the training data generator for the counting visual model or foundation model training.
5. In the main comparision results of Table 1, the listed agentic baselines are not included the iterative refinement mechanism with feedback.

**Audience:**

Yes

**Audience Explanation:**

Yes, researchers who work on high-instance image generation may benefit from this refinement-loop agentic pipeline.

**Claims And Evidence:**

Yes

**Claims Explanation:**

The paper includes a detailed method introduction and a comprehensive comparison with other baseline methods, showing clear performance improvements.

**Requested Changes:**

The requested changes:
- A clear illustration of the Parameter-Free Refinement module. If authors can provide a generation case with full multiple feedback messages and the results of each iteration, it would be very helpful to understand what happened during the refinement loop.

- the reply to weaknesses above.

---

### Review · Reviewer_cwSv · 2026-07-20

**Summary Of Contributions:**

Summary:

This paper presents CountLoop, which is a framework for high-instance text-to-image generation - i.e., prompts that specify a large pricise object count. This is a problem where existing image generation models are known to saturate or miscount, and have attribute binding problems. The proposed framework has three stages. First the input prompt is parsed into a planning graph encoding object attributes and spatial relationships. Second, it goes through a layout-conditioned image generation process based on the planning graph. Third, a VLM critic model then evaluates both the spatial coherence & appearance fidelity and the counting accuracy. The entire process is iterative until the quality criteria are met. Specifically, to mitigate semantic leakage a cumulative attention mechanism is proposed.

Strengths:
1. The problem is well-motivated
2. The evaluation is conducted comprehensively on two benchmarks and also qualitatively.
3. Implementation details are clearly presented.

Weaknesses:
1. The major contribution is the entire critic loop and iterative paradigm, which not entirely new, but very reasonable though.
2. The computational cost is not detailed clearly.

**Audience:**

Yes

**Audience Explanation:**

The problem that this method tries to solve is meaningful and unresolved in the community. The proposed approach is generally architecture-agnostic and could reasonably generalise to future stronger text-to-image generation model.

**Broader Impact Concerns:**

N.A.

**Claims And Evidence:**

Yes

**Claims Explanation:**

Both qualitative and quantitative results are provided. Any further explanations needed are detailed in the "Requested Change" section

**Requested Changes:**

1. The computational cost: The cumulative latent composition method - does it require linear complexity w.r.t. the object count, which could be quite slow during inference? The authors could provide more details and explanations.

2. I believe it's good to diagnose the contribution of each part of the framework, particularly I am interested to these ablations:
2.1. - The functionality of the critic pipeline - How is the performance "without applying the VLM critic pipeline performance" (i.e. single forward run) compared to the layout-to-image generation methods?
2.2. - What if applying the existing layout-to-image methods together with the proposed VLM critic pipeline?

2. How well does the generated image follow the intermediate planning graph? What if the wrong step is happening at the layout aligned generation step, instead of anything wrong with the planning graph generation stage or critic stage. How does the feedback signal from the critic VLM effectively improve this part? Is it simply randomly try multiple times until reaching a correct solution. Correct me if I'm understanding the iterative refinement process wrongly.